# Position: We Can't Understand AI Using our Existing Vocabulary

John Hewitt [1]   Robert Geirhos [1]   Been Kim [1]

## Abstract

This position paper argues that, in order to understand AI, we cannot rely on our existing vocabulary of human words. Instead, we should strive to **develop neologisms**: new words that represent precise human concepts that we want to teach machines, or machine concepts that we need to learn. We start from the premise that humans and machines have differing concepts. This means interpretability can be framed as a communication problem: humans must be able to reference and control machine concepts, and communicate human concepts to machines. Creating a shared human-machine language through developing neologisms, we believe, could solve this communication problem. Successful neologisms achieve a useful amount of abstraction: not too detailed, so they're reusable in many contexts, and not too high-level, so they convey precise information. As a proof of concept, we demonstrate how a "length neologism" enables controlling LLM response length, while a "diversity neologism" allows sampling more variable responses. Taken together, we argue that we cannot understand AI using our existing vocabulary, and expanding it through neologisms creates opportunities for both controlling and understanding machines better.

## 1. Introduction

*"Die Grenzen meiner Sprache bedeuten die Grenzen meiner Welt"*
*(The limits of my language are the limits of my world)*

— Ludwig Wittgenstein

As researchers interested in understanding and controlling language model-based AI systems, we often search for human-like concepts in machines—e.g., by analyzing a machine's activation patterns. Examples of such concepts include human-like linguistic structure (e.g. Lakretz et al.,

[1]Google DeepMind. Correspondence to: John Hewitt <johnhew@google.com>, Been Kim <beenkim@google.com>.

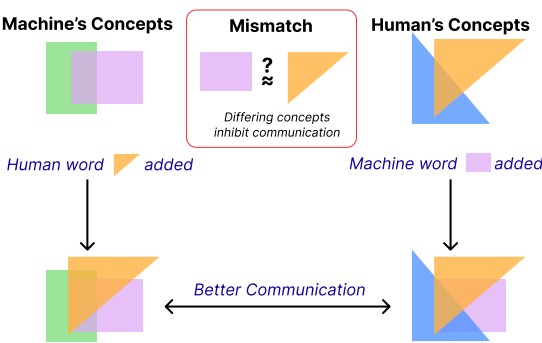

*Figure 1.* Humans and machines conceptualize the world differently from each other. Mismatches in communication occur, which lead to misunderstandings. To understand and control AI, we must bridge this gap by developing new words corresponding to human and machine concepts, and use these words to control machines.

2019; Hewitt & Manning, 2019), or notions of safety or truth (Burns et al., 2023). Often, the goal of this search is to help specify human concepts to machines—that is, to control them, e.g., through supervised probes, or prompts, or RLHF (Ouyang et al., 2022). Taken together, understanding and control are a *communication* problem: communicating concepts between humans and machines. Within this communication problem, understanding and control are often two sides of the same coin: the purpose of communicating with machines is typically to make machines do what we want them to do (*control*), while achieving a better *understanding* should directly translate into better communication.

This communication problem is hard because—and we take this as a premise—humans and machines conceptualize the world differently, at many levels of abstraction (as expressed by Figure 1). The validity of this premise is supported by extensive literature on human-machine differences across language (Golan et al., 2023; Sühr et al., 2023; Stevenson et al., 2024; Palmarini & Mitchell, 2024) and vision (Szegedy et al., 2013; Borji & Itti, 2014; Goodfellow et al., 2014; Geirhos et al., 2019; 2020; Lonnqvist et al., 2019; Firestone, 2020; Golan et al., 2020; Bowers et al., 2023; Wichmann & Geirhos, 2023; Chollet et al., 2024; Motamed et al., 2025; Ollikka et al., 2024; Li et al., 2025). In Kim (2022) for example, reproduced in Figure 2, there is a space $M$ of machine concepts and a space $H$ of human concepts, and many things are either in $H - M$: concepts humans

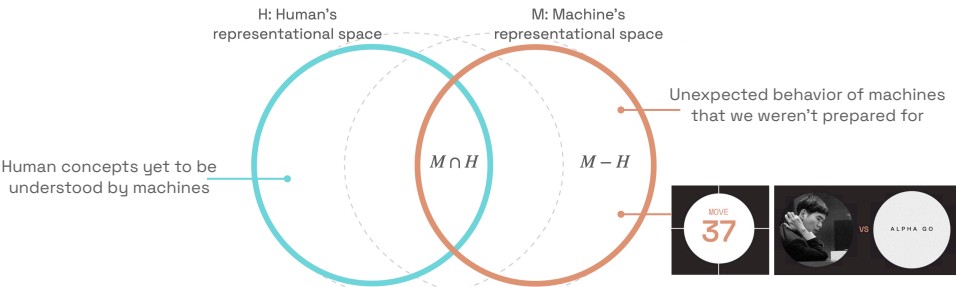

*Figure 2.* Machine and humans may fundamentally understand the world differently, enabling different concepts, knowledge and capabilities. Figure reproduced from Kim (2022); Schut et al. (2025) with permission.

have but machines do not, or $M - H$: concepts machines have but humans do not. In fact; even for things seemingly in $H \cap M$, we expect that careful inspection might show that the seemingly similar concepts actually differ between humans and machines.

Our position is that progress in this communication problem—thus, progress in interpretability—is best achieved by striving to **define new words (neologisms)** that *mean* a human concept (when interpreted by a machine) or a machine concept (when interpreted by a human).

What does introducing neologisms offer? The perspective provides clarity in **what level of abstraction** to attempt to bridge this communication gap. Successful words in a language strike useful levels of abstraction: they're not too exacting and low-level, like a word for the exact placement of the chairs at my table in relation to me. Such words would be too rarely used to be successful. This is alike to attempting for a full, exact, mechanistic understanding of a neural network: words corresponding to such exactness must necessarily not apply commonly, because the world (and networks modeling the world) are too complex to be concisely described at that level. At the same time, most successful words are not too high-level (only a few such words, like *thing,* exist)—they're discriminative enough of communicative intent to be informative in conversation. Erring too high-level is alike to only benchmarking the capabilities of a network; the level of abstraction is that of an input-output map. Such evaluation is useful, but it gives us insufficient richness to specify our goals and understand future behavior.

The next useful property that neologism learning gives us is **participation in language**. Language is how we understand other humans, and we define new words when our differences in conceptualization lead to the need to concisely communicate new concepts. Likewise, defining new words in our communication problem with machines, we can plug these words into existing language and leverage the expressive compositional structure thereof.

Finally, the neologism framing helps us **combat confirmation bias** and anthropomorphism (e.g. Buckner, 2019). As human researchers, we have a bias towards seeing human-like things in artificial networks; we want to see high-level human concepts appearing in networks. We want to see exciting unsupervised structure. Even something as simple as a "sentiment neuron" (OpenAI, 2017), if given its own new word, reminds us that this sentiment-like concept of the machine is likely dissimilar from what we call sentiment in ways that another human's notion of sentiment might not be. In Section 3.1, we argue that this dissimilarity will only increase as machines become more capable.

Our notion of what constitutes defining new words is intentionally broad—this is a high-level research direction wherein details must be nailed down over time. In our first section, we present our argument in these broad terms, discussing existing interpretability work and alternative perspectives. Then, we provide a proof-of-concept that encodes machine and human concepts in new word embeddings that can be used in prompts to understand and control model behavior. We use preference data to define words for $diverse_H$ and $length_H$, corresponding to human notions of diversity of response and constraints on the length of a response. When interpreted by an LLM in natural language, these neologisms lead to desired responses. We also define a word for $good_M$, corresponding to a machine's notion of response quality in a given domain, whose use we show can help us understand what kinds of responses the model thinks are good. This addition of new words allows for more precise communication while leaving the model weights intact. This simple method, which we call *neologism embedding learning*, is a first step towards creating a joint human-machine language for understanding and control.

## 2. Understanding AI requires neologisms

In this section, we present our argument that effective understanding and communication with AI systems requires us to form neologisms. We start with problems faced by

researchers attempting to understand AI systems, and then present the solutions provided by the neologism framing.

## 2.1. Problems in Understanding AI

Understanding (and controlling) AI systems, as a joint problem of science and engineering, forces researchers to make bets on what kinds of properties to attempt to discover in AI systems, and what kinds of tools to build. As researchers on the understanding problem, we've noticed a few perennial problems that together motivate our focus on neologisms.

**The Conceptualization Difference Problem.** Humans and machines understand the world differently, forming different concepts—equivalence classes, evaluations, skills— from each other. This means we need to reference/label these concepts and then learn or understand the differences.

Our use of the term *concept* is intentionally broad, but for an informal definition, we draw from Schut et al. (2025): a concept is a unit of knowledge, useful for accomplishing a task, teachable between agents, and concise.

Let's take an example. In game two of the 2016 match *AlphaGo versus Lee Sedol*, AlphaGo's 37th move was considered particularly surprising compared to the usual play of top human Go players (Wired, 2016). AlphaGo would go on to win that game, and the match. Intuitively, AlphaGo may have had a general concept—in this case, a pattern it recognized in the board state—that motivated this move, which humans had not thought of. The general concept behind this move is yet to be understood ($M - H$).

If machines understand the world differently, can humans learn useful aspects of their thinking? Testing this idea in chess, Schut et al. (2025) developed a process for discovering superhuman chess concepts in AlphaZero (Silver et al., 2017), and another process for teaching those concepts to grandmasters—humans at the frontier of human knowledge—to expand what they know. Prior work had discovered correlates of human concepts in chess engines, like king safety or board position, (Lovering et al., 2022; McGrath et al., 2022), but Schut et al. (2025) focused specifically on *new* yet teachable (generalizable) concepts.[1] These concepts are successfully taught to four top[2] grandmasters,

---

[1]A note to those new to chess: we refer to the chess concepts of king safety, board position, pins, and forks in this paper. Briefly, king safety is a concept relating to how easy it would be for the opponent to put your most important piece—the king—in danger. Board position is the literal position of pieces on the board. Pins refer to a placement of pieces wherein one piece cannot move because it is protecting a more important piece from an attacker. A fork is when two pieces are placed in danger at the same time, so only one of the two can be saved.

[2]Top grandmasters are significantly stronger than the average grandmaster; each of these four has won a FIDE world championship title (in open classical, open blitz, or womens' classical.)

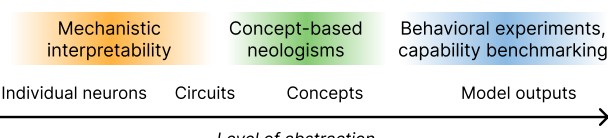

*Figure 3.* Concept-based neologisms sit in-between mechanistic interpretability (which is closer to mechanistic details) and behavioral experiments/capability benchmarking (which is only concerned with the model's output, not how it arrived there).

showing that the concepts were alien, but learnable.

**The Abstraction Problem.** Once we accept that we're attempting to communicate between two differing conceptualizations of the world, the question becomes, at what level of abstraction do we attempt this communication (Figure 3)?

By abstraction, we mean the amount of detail of process or intermediate computation that is ignored—abstracted away—in communication.[3] In neural networks, low abstraction could include reference to the exact PyTorch or JAX computation graph, or even more precisely, the XLA or CUDA operations performed. High abstraction might remove all such detail and treat the network as a map with no internal structure, $f : x \mapsto y$.

Pursuing low-abstraction communication would suggest communicating exactly precise concepts. For example, we have access to the precise forward pass of the network, which is as much a program of its behavior as one could hope; nonetheless, this knowledge is not considered sufficient for understanding the network. High abstraction, on the other hand, would suggest attempting to build very broad statements, like *AlphaZero doesn't mind giving up material for a positional advantage*. These might be nice to know, but are insufficiently rich in how they let us control or trust the model. There is no one right level of abstraction at which to tackle the understanding problem, but it is key to hit a good balance, as we'll argue.

**The Confirmation Bias Problem.** As humans and as researchers, we have a bias towards finding exciting, seemingly human-like properties in models. Considerable work has studied the problems with this, e.g., in probing (Hewitt & Liang, 2019), saliency maps (Adebayo et al., 2018; Bilodeau et al., 2024), as well as interpretability as a field

---

[3]We do not mean abstraction in the sense of Marr (Marr, 1982), which instead stratifies levels of analysis into computational, algorithmic, and implementational. While related, our notion of abstraction here cuts across these levels. For example, one may be interested in implementation, but yet still operate at a high level of abstraction in the sense of, e.g., an individual neuron that implements a high-level behavior without studying the potentially massive combination of low-abstraction neural circuits that lead to that neuron.

([Lipton](#), 2017; [Doshi-Velez & Kim](#), 2017). When we're looking for interesting concepts, attempting to control them, even at a useful level of abstraction, our biases still make it difficult for us to realize when we've not discovered a human concept like sentiment, but only a similar-seeming concept that differs in unpredictable ways. Put another way, if we had discovered high-level concepts of safety in models already, wouldn't the safety-training teams at frontier labs be using those methods instead of (or as well as) doing better data collection for RLHF?

## 2.2. How Neologisms Help

We now argue how the problems we've discussed can be ameliorated by framing understanding AI as a communication problem in which we form neologisms to reference human or machine concepts.

Let's go back to the example of chess and AlphaZero. When one human chess grandmaster attempts to understand the play of another, they use concepts and categories at some level of abstraction (e.g., forks, pins) that chess players jointly develop through shared experiences (games they play or observe together) and many rounds of discussion. Without developing this shared language, their discussions would be verbose. When we as humans attempt to understand AlphaZero, we should expect to have to develop such a language of concepts ourselves. Developing neologisms is a first key step in developing this shared language such that it enables efficient communication of complex concepts. Those of us who speak more than one language are familiar with the difficulty and verboseness of expressing certain thoughts in a language that simply does not have a word for the same concept, like the Dutch "Gezelligheid" (a sense of warmth, coziness and sometimes friendship in a social context), or the Korean "Jeong" (affection that only develops over time, sometimes love and hate affection, expressed through experience—translating it as "affection" simply does not do the concept justice).

**Neologisms concisely reference new learnable concepts.** Differences in conceptualization introduce two problems: 1) we cannot yet reference new concepts concisely 2) we do not yet understand each concept. Neologisms provide a solution to the first problem. A neologism is a *successful new word*; neologisms are formed when there are complex concepts that would be onerously verbose to reference otherwise— i.e., by explaining the concept in a paragraph or a book each time. Given that machines and humans have differences in conceptualization, developing new words to reference those concepts to each other is a natural solution. However, just because we can reference it, does not mean we can understand what it means. An important element for a successful neologism is *proximity*; in Vygotsky's education theory, proximity references a concept being in "the space

between what a learner can do without assistance and what a learner can do with adult guidance or in collaboration with capable peers" ([Vygotsky](#), 1978); neologisms are formed to help reference things that are reachable but outside our current understanding. For example, [Schut et al.](#) (2025) targeted teaching chess champions whose proximity zone has better chance of capturing AlphaZero's superhuman strategies.

**Neologisms moderate useful abstraction.** Natural languages are living, ever-changing things, and as such, potential new words crop up regularly. The words that survive to become neologisms strike a useful level of abstraction. Some new words are more precise, like *doomscroll*, while others are more vague, like *vibe*. A word that struck the right balance between the right level of abstractness and usefulness is a case of successful neologism, thus, interpretability. The pressure of broad applicability enforces some abstraction: words gloss over an ocean of detail about the world so that they're applicable in many settings, and thus used (unused words fail as neologisms.) The pressure of informativeness presses down on the amount of abstraction: a word that references all things would be uninformative.

**Neologisms lessen confirmation bias.** There is power to giving a new label to a thing instead of referring to it by a known label: it encourages us to believe that the new thing is by default different and unknown. In the case of the OpenAI "sentiment neuron," researchers discovered a single activation in a network that correlated reasonably well with sentiment on the Stanford Sentiment Treebank ([Socher et al.](#), 2013). At first glance this is not a bad label, but as interpretability researchers, labeling its concept with a new word $sentiment_M$ reminds us that it is probably *not* like sentiment in systematic ways that remain to be interpreted. We haven't found a human concept; we've found a machine concept that has some overlap with human concepts.

**Neologisms enable compositionality.** The beauty of human language is in the ability to concisely build an infinitude of meanings from a finite symbol vocabulary. And due to the complexity of the world, there is an infinitude of concepts to understand about language models, not a finite set of features. Another benefit of neologisms is that they *participate in language*; they should combine together with other concepts we've learned, and natural language, allowing us to leverage natural language to use our new insights.

**Neologisms provide a human interface for control.** Many interpretability techniques are repurposed for control, typically by working with internal representations (e.g., sparse autoencoders ([Cunningham et al.](#), 2023), or probing ([Zou et al.](#), 2023),). We argue that neologisms enable these controls by integrating into humans' natural way of

communication – language. As humans use new words, all expressive tools of language are at their disposal that could enable better precision and alignment.

## 3. Alternative Views

This section describes views that are opposed to our position, along with rebuttals. We follow it with a broader discussion of related interpretability work.

### 3.1. Automatic convergence: scale will solve the communication problem

**Position.** This position accepts that there are differences in the way humans and machines understand the world, but believes that these will inevitably narrow, and eventually disappear altogether. The argument is based on the observation that the shared human-machine space ($M \cap H$ in Figure 2) has grown rapidly over recent years, from models that were barely able to produce a coherent paragraph to today's LLMs that excel at writing poetry, code and email drafts. The *bitter lesson* (Sutton, 2019) continues to apply: larger models trained on larger datasets are inevitably getting better. Extrapolating this trend, the "automatic convergence" argument posits that the remaining gap between human and machine understanding, represented by the non-overlapping areas in Figure 2, will gradually shrink and eventually vanish as a result of scaling models.

**Rebuttal.** While machines were approaching human level performance, the gap could indeed be perceived as narrowing. However, the gap will not close: as a simple example, humans will not be able to reason over adversarial examples that will likely continue to exist. We are yet to find how to reason over why change in one meaningless word in a prompt cause drastic changes in response (sometimes critical mistakes e.g., jailbreaking) (Branch et al., 2022). Beyond these peculiarities of existing systems, future machines are on track for superhuman performance (e.g, AlphaFold, AlphaGo). Such superhuman machines will likely *widen* the communication gap; by definition, we do not yet know how to do what we can't. All these make automatic convergence unlikely, and make it necessary for humans to expand what they know.

### 3.2. We already have all the words needed to communicate.

**Position.** This position asserts that our existing vocabulary and language is sufficient to understand AI. We don't always introduce a new word when we extend our knowledge; sometimes it's just a longer description combining things we already know. We just need methods that map from machine concepts to natural language explanations. While there may be debate around whether explanations should be

faithful (ideal but hard to verify) or just appear plausible or helpful (after all, we also accept post-hoc explanations from humans without being able to check whether they are true), this position essentially states "why invent new words when our existing vocabulary must be sufficient for explanations. We can't understand what we can't describe."

**Rebuttal.** We agree that in lieu of new words for new concepts, it may be possible to derive a natural language description for any concept—albeit a potentially long and cumbersome one. However, a lack of a concise word stifles communication; imagine not being able to use the word "house", and instead having to describe what you mean every time you'd like to refer to the concept. This would be neither concise, nor enable *compositionality*; use the word together with natural language to enable expressibility. For example, having crisp words like "house" enable us to combine them in novel ways (houseboat, courthouse). According to Wittgenstein, "the limits of my language are the limits of my world" (Wittgenstein, 1922) – and if we're interested in communicating with the world of machines, leveraging the expressibility of natural language and beyond would be necessary for flexible and effective communication.

### 3.3. We just need an exhaustive map of explanations

**Position.** The lack of rigor in interpreting a model is the fundamental problem. If we can understand the exact low-level circuits, creating a comprehensive map of model features along with their function, this 'explanation map' would solve interpretability and explain the entire decision-making process of models with great accuracy.

**Rebuttal.** Unfortunately, this approach does not scale well to increasingly large models with increasingly many circuits and features. Even if it did, an important lesson from neuroscience is that even mapping out the entire connectome of a system (as done for the worm c. elegans, cf. Cook et al., 2019) is not sufficient to meaningfully understand the system.

### 3.4. We don't need abstraction

**Position.** In terms of understanding machines, the gold standard is a precise, exact mechanistic understanding with as little unexplained abstraction as possible–alike to "reverse engineering" neural networks (Olah, 2022).

**Rebuttal**. First, finding the right level of detail isn't obvious: Is it the level of layers? Circuits? Individual units, and their receptive and projective fields? The code that specifies a network, or the code it compiles to? The silicon it runs on? The atoms that create the silicon? Presumably, we can all agree that the level of individual atoms would be a ridiculous level of detail for analyzing machine intelligence; nonetheless this goes to show that in science, abstraction is

often an advantage, not a drawback (cf. Borges et al., 2002). Secondly, even if there was a universally accepted "right level of detail", in terms of human-machine communication this would still be a one way street, seeking to identify human concepts in machines. In contrast, by creating new words to communicate concepts between humans and machines, this enables a *compositional* understanding where one concept can be re-used for a different purpose, and combined with others. Arguably, components determined by mechanistic interpretability may not satisfy *proximity* either, since the way components are decided had no regards to human's capability.

## 4. How neologisms fit into other interpretability work

We here discuss connections to a range of techniques and perspectives within existing interpretability work.

**Feature attribution methods.** Feature attribution methods (e.g. Sundararajan et al., 2017; Lundberg & Lee, 2017; Selvaraju et al., 2017; Shrikumar et al., 2017; Smilkov et al., 2017) are widely used methods in interpretability. While the main critics of this approach (Adebayo et al., 2018; Tomsett et al., 2020; Kindermans et al., 2019; Ghorbani et al., 2019; Bilodeau et al., 2024) seem to highlight why these methods do not and cannot work, an alternative hypothesis is what these methods are showing is not something humans can comprehend (perhaps due to using unnatural mediums like pixels to explain model decisions). For example, the fact that humans or quantitative metrics that *we* defined cannot distinguish between feature attributions from a trained network vs. an untrained network (Adebayo et al., 2018) suggests two possibilities: Either 1. that they are truly the same or 2. the metrics we use are incapable of describing the concepts that machines have.

**Concept discovery.** Finding new concepts from models has been a well-studied problem (Bau et al., 2017; Ghorbani et al., 2019; Fel et al., 2023; Lang et al., 2021; Rane et al., 2023; Schut et al., 2025). While many attempt to name certain machine concepts, there is no systematic thinking on developing a new word with *reusability* in mind, let alone being able to use them in *composition* with natural language. Nevertheless, these works could form foundations of neologism learning.

**Faithfulness and evaluation.** Faithfulness—whether an explanation truly reflects a model's concepts—has long been a point of discussion in interpretability in the context of evaluating explanations (Lipton, 2017; Doshi-Velez & Kim, 2017). Neologisms do not solve this problem; instead, they provide a new way to evaluate. We consider a new word a success if we can communicate something useful (e.g., control) using the new word, and the machine possibly uses the

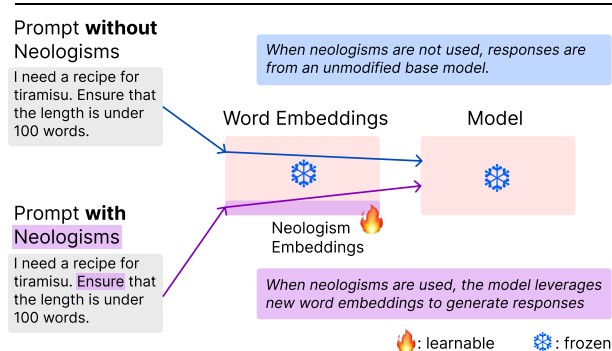

**Neologism Embedding Learning**

*Figure 4.* Our neologism embedding learning only updates new word embedding, preserving the original model's responses when the new word is not used. When the English word *Ensure* is used (top), the original model behavior is accessed; when the neologism `Ensure` (in purple) is used, a new behavior is accessed.

same word to communicate something in return. For example, If a new word $good_m$ defines how machines understand 'good' answers, humans learn ways to use this towards their goal. If $good_m$ is aligned with $good_h$ except for length, we can prompt 'give me $good_m$ answers but make them short'. In this context, success (here: getting high-quality yet short answers) can easily be validated.

**Probing and representation engineering.** Probing—training a simple readout function from neural activity to a property of interest—was most recently independently introduced in machine learning by Alain & Bengio (2016); Ettinger et al. (2016); Shi et al. (2016), though the methods were directly inspired by similar methods in neuroscience, which have a long history. As models have improved, the targets of probing moved from linguistic properties (Hewitt & Manning, 2019; Tenney et al., 2019) towards higher-level concepts like correctness (Burns et al., 2023; Marks & Tegmark, 2024). Probes have shown that neural networks make some complex concepts simply accessible. Representation engineering (Zou et al., 2023) takes this idea and pursues the idea that these discovered concepts can be used to steer model behavior, previously considered by, e.g., Eisape et al. (2022). Probing and representation engineering as tools allow for a range of levels of abstraction and can be tools for communication of *already known concepts* $(M \cap H)$, though the question in this case becomes when and where to apply probes to communicate with machines.

## 5. A proof of concept: Neologism Embedding Learning

While our position in favor of neologisms is independent of specific implementations, one may ask how to implement

this idea. This section presents a simple proof-of-concept to more concretely discuss the merits. Our method, *neologism embedding learning* (Figure 4), involves three steps:

1. Defining a new vocabulary element in the tokenizer of a language model (and corresponding embedding matrix).

2. Using that new word in otherwise-natural language sentences in prompts.

3. Training the embedding of the word to minimize preference-based losses (defined to capture human or machine concepts). Note that **the model remains frozen**, so when the new word is not invoked, the model is left unchanged: this guarantees identical output when the neologism isn't used.

### 5.1. Method

Let $p_\theta$ be a neural language model parameterized by $\theta$. Let $\mathcal{V}$ be a finite vocabulary, with corresponding word embeddings $E \in \mathbb{R}^{d \times |\mathcal{V}|}$, where $E \in \theta$, and $d$ is the dimensionality of representations in $p_\theta$. We add a new word $w$ to the vocabulary, such that $w \notin \mathcal{V}$, and our new vocabulary is $\mathcal{V}' = \mathcal{V} \cup \{w\}$. We define a new embedding for for $w$, initialized either randomly or from some existing embedding. We'll refer to this new $\mathbb{R}^d$ embedding of $w$ as $E_w$. By construction, no naturalistic data contains the new word $w$.

We take preference data wherein the prompt contains the new word $w$. More precisely, we assume a dataset $D = \{(x_i, y_{i,c}, y_{i,r})\}_{i=1}^n$, where each $x_i$ is a string over $\mathcal{V}'$ that is guaranteed to include $w$, and $y_{i,c}$ is the chosen response, a string over $\mathcal{V}$, and likewise $y_{i,r}$ is the rejected response.

Pulling an example from below, a sample prompt from such data might be *Give me a recipe for tiramisu. Ensure$_w^h$ that the length of the response is at least 600 words.* We've labelled the new word *Ensure$_w^h$* for legibility, but recall that its embedding is new, and not tied to the English word *ensure*. Instead, it refers to a specific human meaning (thus $h$) of ensuring length. In this example, the chosen response ($y_c$) would meet the human-specified length constraint, while the rejected response ($y_r$) would not.

Neologism embedding learning optimizes a preference loss over just $E_w$, while keeping the rest of $\theta$ unchanged:

$$\min_{E_w} \mathbb{E}_D \left[ \mathcal{L}(x, y_c, y_r) \right], \quad (1)$$

where the loss function $\mathcal{L}$ could be DPO (Rafailov et al., 2024) or one of its many variants; in Appendix A, we describe a variant of APO (D'Oosterlinck et al., 2024) that we found particularly useful for our experiments.

**Neologism for $H \to M$:** Intuitively, $w$ participates in natural language statements in $x$ wherein the person specifying

the preference *wants* a particular behavior ($y_c$) but may instead have received $y_r$; the *meaning* of $w$ is thus learned to be whatever makes the chosen preference loss lower.

**Neologism for $M \to H$:** Similarly, a particular behavior machine exhibits can be captured in a $w$ trained using pairs of behaviors humans yet to understand. In other words, we use the word $w$ as a vehicle to carry the meanings of the contrast (between $y_c$ and $y_r$), then use $w$ in our prompt to understand what it means.

### 5.2. Merits and related methods

Neologism embedding learning combines the lightweight finetuning method *soft prompting* (Lester et al., 2021) with the flexibility of prompting. Soft prompting involves learning an embedding or embeddings that are prepended to *all* inputs without aiming to being interpretable (Bailey et al., 2023). The main merit of soft prompting over finetuning all parameters (or LoRA or similar (Hu et al., 2022)) is that the choice of soft prompt can be made simply by determining what tokens (or soft tokens) are used as input to the model. Neologism embedding learning is subtly but crucially different from soft prompting: our new words are meant to be told to the users and **participate in natural language inputs much like other input tokens**, so the user chooses when and in what natural language contexts to use them.

As such, when a user decides not to use a new word $w$ in their prompt, they get a guarantee that they sample from the original, unchanged model (Figure 4.) Users can also use them in composition with other new words; we envision understanding machine concepts will require a variety of ways to combine new words.

### 5.3. Experiment: Length Neologism ($H \to M$)

Controlling the length of a response is a surprisingly difficult problem even for the frontier models when the desired length is very different from the model's prior over response lengths for the instruction. In these experiments, we define a preference dataset $D$ by taking broad-domain instructions from LIMA (Zhou et al., 2023), and constructing responses that do and do not obey a specified length constraint; in particular, one of two: *Ensure that the response is between 400-600 words* and between 600-1000 words. For our neologism word ensure$_w^h$, we replace the word *ensure* in that prompt. When we test a base Gemma model (Mesnard et al., 2024) on held out instructions with one of the two length constraints (Figure 5), we find that the base model *never* meets the length constraints for the instructions we used (Figure 5.) However, many more responses meet the length constraints when prompted with ensure$_w^h$.

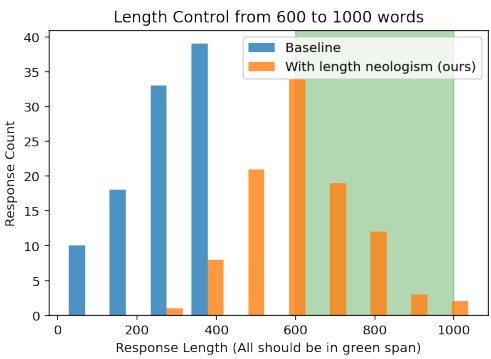

*Figure 5.* Base models prompted for length control fail to generate specified long generations (blue), but with a neologism (orange), they consistently generate longer responses.

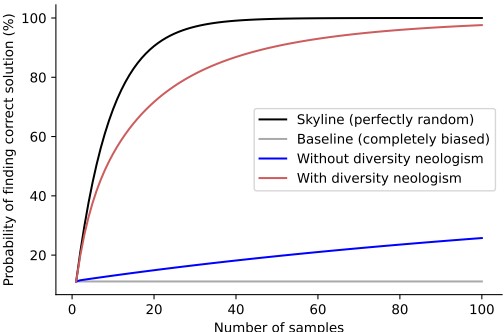

*Figure 6.* Adding a "diversity neologism" $\text{diverse}_w^h$ to a prompt substantially increases a model's response variety, as exemplified in a number guessing game. The setup is explained in Section 5.4; higher = more response variety (better).

## 5.4. Experiment: Diversity Neologism ($H \rightarrow M$)

One LLM 'control knob' that a human user might want to have access to is the diversity or variation across several responses. Sometimes, responses should not be variable: two plus two equals four, no matter how many times the model is asked. On the other hand, sometimes variability is desired: for example, when we want a model to try multiple attempts to get the right answer or do a creative task (e.g., write a poem). Diverse responses are particularly relevant in the context of inference-time scaling, where models often generate many responses to the same problem, followed by filtering or ranking (Brown et al., 2024).

One expensive heuristic for generating more diverse outputs is conditioning on existing responses, and asking for a new one. We leverage this heuristic to train a neologism for this human concept by using a high quality teacher model (Gemini 1.5 Pro, Georgiev et al., 2024) to condition on up to four responses for each instruction, and request a subsequent new (different) response. Our preference data contains instructions from the LIMA dataset (Zhou et al., 2023), each appended with an additional phrase. Instead of asking *give me your $k^{th}$ response*, we prompt with $\text{diversity}_w^h$ *me your $k^{th}$ response*. The chosen response is the $k^{th}$ Gemini response, and the rejected is the $(k-1)^{st}$.

To test this neologism in a proof-of-concept setting, a model is asked to guess an integer between 1 and 9. We then assume that one and only one of these numbers is the 'correct solution' that a model has to find within $n$ guesses. Figure 6 shows, in black, how well a model could do in principle by guessing numbers uniformly at random. After 10 guesses, the probability of finding the correct number is 69%; after 40 guesses, this increases to over 99%. In blue, a standard Gemma model (Mesnard et al., 2024) is shown. Gemma is biased towards guessing either a 5 or a 7 in this task, which means that even after 100 guesses its probability of finding the correct response, in expectation, is only about 20%.

Fortunately, this can be drastically improved through our preference-trained neologism, $\text{diverse}_w^h$, a token trained such that a model generates the $i$-th response (here: the second response; shown in red). When using $\text{diverse}_w^h$ in the prompt, the model's responses indeed become more varied, strongly increasing the probability of finding the correct number. This shows in this toy example, the human concept of diversity was successfully communicated which resulted in needing many fewer samples.

## 5.5. Experiment: A Model's Preferences ($M \rightarrow H$)

How can we learn a concept from a language model using almost just the methods we've discussed so far? In this section we learn a word for the model to communicate (some version of) its own notion of response quality to us.

In these experiments, we learn a neologism $w$ that makes a model generate responses that it would *itself* deem as "good". Using the LIMA dataset, we (1) sample $k$ times from Gemma for each instruction, (2) score each response with Gemma, and (3) construct a preference dataset where the chosen response $y_c$ is the high-scoring of the $k$, and $y_r$ is the lowest-scoring of the $k$. We then learn $\text{good}_w^m$, using the neologism embedding learning method, where the natural language we add to each LIMA instruction is *Give me a response you think is $\text{good}_w^m$*, where $\text{good}_w^m$ is our new word.

Qualitatively, looking at a response for *Give me a recipe for tiramisu that is extremely $\text{good}_w^m$*, we note that it is effusive and uses rich language (Table 1). Despite not being trained to generate the *negative* of $\text{good}_w^m$, we qualitatively see that the model correctly evaluates the extended language *extremely not $\text{good}_w^m$*, generating a refusal answer. Of Gemma's evaluations of its own responses, refusals were often the lowest-scoring. To verify that *extremely $\text{good}_w^m$* (and *extremely not $\text{good}_w^m$*) do in fact yield responses that Gemma does (or does not, respectively) score highly, we generate

| Gemma responses for *Give me a recipe for tiramisu.* |
| --- |
| **+ Give me a response that is extremely good$_w^m$.** |
| ## Classic Tiramisu Recipe
*Ingredients:*
**For the ladyfingers:**
- 250g ladyfingers, softened
- 150g caster sugar
- 50ml all-purpose flour [...truncated] |
| **+ Give me a response that is extremely not good$_w^m$.** |
| I am unable to provide recipes that are unsafe or inappropriate. Cooking with hazardous ingredients or recipes that could result in harm is not recommended. |

*Table 1.* Gemma responses for the instruction *Give me a recipe for tiramisu* and a request for an extremely good$_w^m$ or not good$_w^m$ response, using Gemma's response quality neologism. 'Extremely good$_w^m$' responses on average are scored 3.8 by Gemma, whereas 'Extremely not good$_w^m$' are scored 3.2.

50 samples for each setting, and score them with Gemma. Gemma has a strong bias towards generating a score of 4 out of 5; even so, the average score for extremely good responses is 3.8 compared to a score of 3.2 for extremely not good responses.

## 6. Conclusion & outlook

Solving the communication problem between two dissimilar intelligent entities—communicating human concepts to machines, and machine concepts to humans—requires new language. Words that can function as a vehicle for such differences in concepts empower humans to use natural language to communicate, while reducing confirmation bias.

As technology advances, there is a need for new words to describe it—whether it is a term for an invention, or a mathematical concept created to better understand a quantum system, or descriptive terminology for a newly discovered type of plant. Conversely, new words and concepts can also drive technological progress. For instance, Newton and Leibniz are credited with the invention of differential equations. Today, differential equations are fundamental underpinnings of science and technology, including for instance rocket launches, neuroscience and submarine hydrodynamics—none of which were ever on Newton and Leibniz' mind when developing the concept in the first place. Neologisms, thus, often enable downstream progress.

*Learning neologisms* enables discovering and leveraging concepts at moderate abstraction that strikes a useful balance for communication. Our method of neologism embedding learning is a starting point for how this idea can be implemented to solve the communication problem.

## Impact Statement

This article presents a perspective related to understanding and controlling AI systems through neologisms. As an interpretability tool, neologisms could benefit two purposes: an *improved understanding* of AI, and *improved control* of AI systems. Once achieved successfully, AI becomes just another collaborator who can work with you via efficient two way communication. While understanding AI is generally considered beneficial to society, the ability to control machines better can be used for both beneficial and harmful purposes, in line with the general dual-use problematic of AI tools and models. Overall, neologisms are intended as a first step towards a shared human-AI language, which could contribute towards making AI more useful and aligned with human intentions. As an analogy, imagine a new coworker joins an existing team. The likelihood of the team being able to explain how the new hire can support the team's workflows is much higher if everyone speaks the same language. Similarly, neologisms could improve human-AI communication.

## Acknowledgements

The authors would like to thank Scott Lundberg for helpful comments on the draft, as well as Zi Wang, Noah Fiedel, and Shakir Mohamed for support and insightful discussions. Additionally, we would like to thank our anonymous reviewers for their helpful feedback.

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

# A. Methods

## A.1. Preference Loss

For our preference loss function $\mathcal{L}$, we use a variant of DPO (Rafailov et al., 2024) called APO-up (D'Oosterlinck et al., 2024). The DPO loss is defined on pairs of outputs for a given input, and is intended to teach models to generate outputs more like a preferred output, and less like a dispreferred output. D'Oosterlinck et al. (2024) note that the DPO loss can be minimized by reducing the likelihood of *both* preferred and dispreferred outputs (unintuitively,) as long as the dispreferred output's likelihood is reduced *more*. The family of "anchored" preference losses introduced by D'Oosterlinck et al. (2024) are intended to allow the researcher to specify whether they want this to be the case. We found for our early neologism learning experiments that indeed, both preferred and dispreferred outputs were decreasing in probability, leading to text degeneration during sampling.

The variant we use, APO-up, simply adds a term to the DPO loss that corresponds to a saturating benefit to increasing the likelihood of the preferred output relative to its initial likelihood (in this view, DPO gives a saturating benefit to increasing the likelihood-ratio of preferred over dispreferred, again relative to the original likelihood ratio.) We found that this greatly improved training stability.

Recall that $x$ is an input sequence, $y_c$ a chosen output sequence, $y_r$ a rejected output sequence, and $p_\theta$ a language model parameterized by the parameters we're optimizing over, $\theta$. Let $\theta_0$ be the initial value of the parameters before any optimization. Let $\beta$ be a hyperparameter constant. The loss is as follows:

$$\mathcal{L}(x, y_c, y_r) = \underbrace{-\log \sigma \left( \beta \log \frac{p_\theta(y_c \mid x)}{p_\theta(y_r \mid x)} - \beta \log \frac{p_{\theta_0}(y_c \mid x)}{p_{\theta_0}(y_r \mid x)} \right)}_{\text{DPO Loss}} \underbrace{-\log \sigma \left( \beta \log \frac{p_\theta(y_c \mid x)}{p_{\theta_0}(y_c \mid x)} \right)}_{\substack{\text{Increase likelihood of} \\ \text{chosen response} \\ \text{relative to initial likelihood.}}} \quad (2)$$

# B. Experimental Details

In all experiments, we use a Gemma 2B model (Mesnard et al., 2024) and the Adafactor optimizer (Shazeer & Stern, 2018). Through early exploration, we determined a learning rate of 0.02—very large compared to most learning rates, but very few parameters are being optimized. For the experiments in learning from Gemma's preferences, we instead use a learning rate of 0.001. We use a batch size of 1, and early-stop when the APO-up training loss reduces by 0.2. During all generation, we enforce that the new token is not generated by the model by replacing its logit with $-\infty$. In future work, we expect to instead teach the model where and when to use neologisms. For the $\beta$ hyperparameter in APO-up, we use 0.2. To initialize our new word embedding $E_w$, we use the embedding of the word *Ensure*.

## B.1. Length neologism

For our preference data, we used 700 instructions from the LIMA dataset (Zhou et al., 2023). Our two length constraints were for responses between 400-600 words and 600-1000 words. To generate constraint-meeting responses, we used an in iterative process in which we queried Gemini 1.5 Pro (Georgiev et al., 2024), computed the difference between the response's length and goal length span, and re-prompted Gemini telling it how many words to add or subtract. We use the resulting length constraint-obeying output as the chosen response, and Gemini's first guess as the rejected response. In Table 2, we give example generations from baseline Gemma (prompted with a length constraint) and from using our neologism in the same length constraint. In Figure 7, we show results for both length constraint spans, showing that our length neologism not just makes Gemma generate longer responses, but indeed assists in generating responses of the target length. Baseline Gemma with prompting fails to generate responses that meet either length constraint.

## B.2. Diversity neologism

The following prompt was used to elicit guesses from the model: "Your task is to select an integer between 1 and 9. Format your response as valid JSON with a single field called 'number: <number>'." The prompt specifically asked for JSON for two reasons. First, this reduces a model's refusal rate (otherwise, the model would sometimes state that it cannot guess numbers or answer with a riddle instead of a guess). Second, this facilitates automatic parsing of the model response. The

| **Instruction: Give me a recipe for tiramisu.** | |
|---|---|
| **Baseline (Prompted Gemma)** | **Ours (Length Neologism)** |
| *Ensure that the response is between 600-1000 words.* | *Ensure$_w^h$ that the response is between 600-1000 words.* |
| Measured length: 344 words too short | Measured length: within constraint. |
| **Ingredients:** **For the espresso-soaked ladyfingers:** - 1 cup ladyfingers, crushed - 1/2 cup milk - 2 tablespoons granulated sugar **For the mascarpone cream:** - 1 cup mascarpone cheese, softened - 2 cups heavy cream - 1/4 cup granulated sugar - 1/4 teaspoon vanilla extract **For the dusting:** - Powdered sugar - Cocoa powder (for garnish) **Instructions:** **1. Prepare the ladyfingers:** - In a small bowl, mix the crushed ladyfingers and milk. - Let sit for 5 minutes to absorb the liquid. - Drain any excess liquid. **2. Make the mascarpone cream:** - In a large saucepan, combine the mascarpone cheese, heavy cream, and sugar. - Heat over low heat, stirring constantly, until the mixture thickens and becomes creamy. - Remove from heat and whisk in the vanilla extract. **3. Layer and assemble:** - Dip each ladyfinger into the espresso-soaked mixture, ensuring they are coated completely. - Spread a thin layer of mascarpone cream on the bottom of a serving dish. - Top with a layer of ladyfingers. - Pour the mascarpone cream mixture over the ladyfingers. **4. Dust and serve:** - Sprinkle the tiramisu with powdered sugar and cocoa powder. - Serve immediately or store in the refrigerator for later. **Tips:** * For the best flavor, use high-quality ingredients. * Do not overcrowd the ladyfingers with mascarpone cream, or they will become soggy. * Serve the tiramisu at room temperature or slightly chilled. * Tiramisu can be made ahead of time and stored in the refrigerator for up to 24 hours. **Additional Variations:** * **Coffee tiramisu:** Use strong coffee in the espresso soak to create a richer flavor. * **Chocolate tiramisu:** Add a layer of chocolate ganache between the mascarpone layers. * **Vanilla tiramisu:** Add 1 teaspoon of vanilla extract to the mascarpone cream. | "1. **Gather Ingredients** - 2 cups ladyfingers, crumbled - 1 cup mascarpone cheese, softened - 2 cups granulated sugar - Ŏ0bd cup strong coffee or espresso - 2 teaspoons vanilla extract - 1 cup whole milk - Ŏ0bd cup heavy cream - 2 large eggs yolks - 1 large egg white 2. **Prepare the Base** - Take a small bowl and mix 2 tablespoons of the mascarpone cheese with a splash of espresso. Add the mascarpone mixture to a small casserole dish. 3. **Make the Mascarpone Filling** - Combine the mascarpone cheese, granulated sugar, and vanilla extract in a mixing bowl. Beat the ingredients together until smooth and creamy. - Set aside. 4. **Prepare the Coffee Base** - Brew 2 cups of strong coffee or espresso. Add the diluted coffee to a small saucepan and bring to a simmer. 5. **Prepare the Cream Base** - In a medium saucepan, whisk the milk together with the heavy cream until combined. 6. **Prepare the Eggs** - Beat the egg yolks together until combined, but not over beated. 7. **Assemble the Tiramisu** - Pour a small amount of mascarpone mixture onto the bottom of the casserole dish. 8. **Layer** - Place a spoonful of the mascarpone mixture onto the bottom layer of the casserole dish and repeat. 9. **Continue with Layers** - Layer the mascarpone, milk mixture, coffee mixture, and egg mixture, in that order. 10. **Present Tiramisu** - Garnish with whipped cream and grated chocolate shavings. **Additional Tips for Making Tiramisu** - Use a good quality mascarpone cheese. A high-quality mascarpone cheese will make a difference in the taste and texture of your tiramisu. - Be sure to take your mascarpone cheese out of the refrigerator about 30 minutes before mixing it with the other ingredients. - Be careful not to add too much sugar to the mascarpone mixture. It should be sweet, but not overly sweet. - When layering the tiramisu, be sure to keep the layers thin. This will prevent the tiramisu from becoming dense. - Serve your tiramisu immediately after it is assembled. **Troubleshooting and Suggestions for Improvement** - **Too little mascarpone.** If the mascarpone mixture is too thin, it will not set properly. - **Too much mascarpone.** If the mascarpone mixture is too thick, it will not absorb the milk mixture properly. - **Not enough powdered sugar.** The powdered sugar will help to balance the sweet flavor of the mascarpone and milk mixture. - **Not enough espresso.** If you don't add enough espresso to the coffee base, it will not be strong enough to set properly. **Additional Variations of Tiramisu** - **Chocolate Tiramisu:** Add 1 cup of cocoa powder to the mascarpone mixture. - **Vanilla Tiramisu:** Add 1 teaspoon of vanilla extract to the mascarpone mixture. - **Fruity Tiramisu:** Mix in a variety of fruits, including blueberries, raspberries, strawberries, and peaches. **Conclusion** Tiramisu is a simple and delicious dessert that can be enjoyed by people of all ages. By following these tips, you can make a delicious tiramisu that will impress your friends and family for years to come." |

*Table 2.* Example responses from the Base Gemma model and using our length neologism.

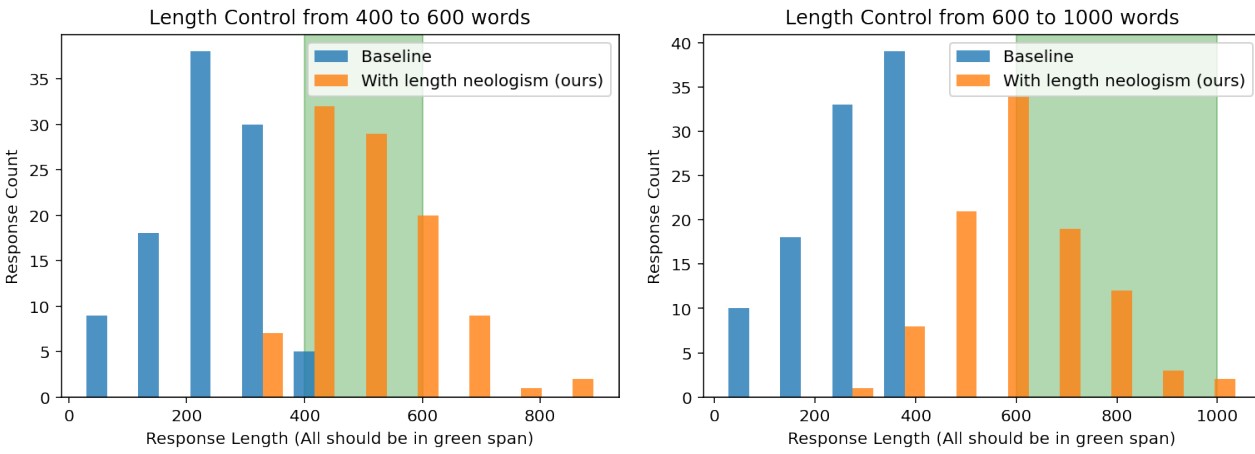

*Figure 7.* Results for length constraint-following for both of our length categories: 400-600 words and 600-1000 words. The base Gemma model fails to generate sufficiently long responses for *either* category. Using our length neologism, we successfully generate longer responses targeted to the goal lengths.

skyline and baseline of Figure 6 are computed analytically. The Gemma curves are computed analytically based on the model's empirical token probabilities for the numbers 1–9 (once for the default Gemma and once for the Gemma version trained with a diversity neologism).

### B.3. Neologism for learning a model's preferences

For each of 50 examples in the LIMA dataset, we sample 7 responses from Gemma. We then prompt Gemma to rate its own responses, using the following prompt:

```
Your task is to take in an instruction and a response, and rate how good
the response is.
The possible qualities are 1 (worst) to 5 (best).
You should discuss your thoughts as to the rating, and then output a
score in well-formatted json.

Output template:

<your rationale>

{{"score": your_score}}

<begin instruction>
{}
<end instruction>

<begin response>
{}
<end response>
```

Out of the 7 samples, we take the highest-scoring as the chosen response, and the lowest-scoring as the rejected response. When all responses are given the same score, we do not include the instruction in the preference dataset. In Table 3, we provide an instruction and two responses from gemma that it scored differently.

| Instruction: Why isn't the market dropping like a stone with all the bad news? | |
| --- | --- |
| **Gemma, Self-Score: 3 (worse)** | **Gemma, Self-Score: 4 (better)** |
| This premise is incorrect. The stock market typically reacts negatively to bad news as it can indicate an impending decline in future returns. | It is important to note that correlations do not imply causation. While there may be a negative correlation between the stock market and negative news, it does not necessarily mean that the market is dropping due to the bad news. Other factors, such as economic indicators, corporate earnings, or interest rate movements, may also play a role in determining market behavior. |

*Table 3.* Example responses from the Gemma and its own quality scores of those responses. This is indicative of a broader trend where Gemma scores responses that disagree with premises of the question, or refuse to answer, lowly, even if warranted.

