# OpenReview forum: "Position: We Can’t Understand AI Using our Existing Vocabulary"
_ICML.cc/2025/Position_Paper_Track — ICML 2025 Position Paper Track spotlightposter_

### Official Review · Reviewer_oDL4 · 2025-03-13

**Significance:** 2
**Argument Clarity:** 3
**Rating:** 3
**Confidence:** 4

**Questions:**

__Q1.__ The authors' argument for the need for abstraction seems to converge with theoretical results on the computational complexity of low-level (i.e., less abstract; neurons, circuits) interpretability queries (see e.g., Bassan et al., 2024; Adolfi et al., 2025; Barcelo et al., 2020). Are higher-level abstractions in this paper proposed as not only more intelligible to humans but also more computationally tractable? If not, what are the possible computational barriers on these higher level abstractions exploited by neologisms?

__Q2.__ Can you make the chess examples accessible to non-chess players? E.g., "forks" and "pins" are potentially unfamiliar jargon.

__Q3.__ Can you clarify the reason why "model outputs" are higher in abstraction level than circuits and concepts? These categories seem incommensurable. Perhaps what the authors mean for the right-most is something like "capacity" (e.g., a capability stated as a formal computation).

__References__

Bassan, S., Amir, G., & Katz, G. (2024). Local vs. Global Interpretability: A Computational Complexity Perspective. Proceedings of the 41st International Conference on Machine Learning, 3133–3167.

Adolfi, F., Vilas, M. G., & Wareham, T. (2025). The Computational Complexity of Circuit Discovery for Inner Interpretability. The Thirteenth International Conference on Learning Representations.

Barceló, P., Monet, M., Pérez, J., & Subercaseaux, B. (2020). Model Interpretability through the lens of Computational Complexity. Advances in Neural Information Processing Systems, 33, 15487–15498.

Vilas, M. G., Adolfi, F., Poeppel, D., & Roig, G. (2024). Position: An Inner Interpretability Framework for AI Inspired by Lessons from Cognitive Neuroscience. Proceedings of the 41st International Conference on Machine Learning, 49506–49522.


**NOTE: The reviewer's assessment is in *no* way tied to the inclusion of the references in the manuscript. These are provided as examples or pointers to the literature.**

**Discussion Potential:**

2

**Paper Summary:**

This paper frames interpretability by analogy to a communication problem between humans and machines and proposes that our current vocabulary is ill suited for this purposes.
The authors propose to develop neologisms and to use them as common ground for this 'communication' (i.e., to understand and control AI).
The paper provides extensive conceptual argumentation for the proposal, discusses related issues and alternative views, and develops a technical proof of concept: a method termed "neologism embedding learning".

## Update after rebuttal

The proposed changes seem adequate. Note, however, that these changes are largely promised in abstract terms and not exemplified concretely. I will therefore keep my score.

**Position:**

Yes

**Position In Title:**

Yes

**Related Work:**

2

**Strengths And Weaknesses:**

__Strengths__

__S1.__ The paper is well written, contains examples, explanations and diagrams, and the arguments are presented in an accessible way.

__S2.__ It addresses issues that are relevant to interpretability/explainability communities.

__S3.__ It provides extensive conceptual support for the position, considers alternative views, and provides examples of how the proposal could be implemented in practice.

__Weaknesses__

__W1.__ Some paragraphs lack engagement with the relevant literature. For instance, "the question becomes, at what level of abstraction do we attempt this communication" (p. 3), this issue has been discussed recently (e.g., Vilas et al., 2024).

__W2.__ At various places, the authors use somewhat vague language without following it up with concrete explanations. An example is "we still fall victim to our own biases when evaluating whether those concepts, like safety or sentiment, really line up with our own". It would be helpful to follow up these statements more concretely or avoid vague language altogether (e.g., "fall victim to").

__W3.__ (p. 4) It is claimed that neologisms "provide a human interface for control" but at this point this is not explained and the relevant section is a paragraph with 3 sentences that mainly states the position. More generally, I would suggest to avoid affirming conclusions unless they are preceded or followed up with arguments and explanations.

__W4.__ There are multiple assertions without support or citation. Some examples include:
(1) "This position accepts that there are differences in the way humans and machines understand the world, but believes that these will inevitably narrow, and eventually disappear altogether."
(2) "humans will not be able to reason over adversarial examples that will likely continue to exist (an optimization will always get us one)."
(3) "This position asserts that our existing vocabulary and language is sufficient to understand AI."
(4) "abstraction is often an advantage, not a drawback". In (4) a citation to a (relevant) literary text is provided but a scientific citation is needed as well (e.g., Vilas et al., 2024)

__W5.__ Figure 4 is a bit confusing to this reviewer. Prompts with and without neologisms are exactly the same, only that one word is highlighted in purple. It is not clear what this is supposed to mean and the figure caption does not help. Perhaps move the key idea in the text to the caption to make the figure more self-contained.

__W6.__ The word "concept" is used but not explained or defined. This is somewhat confusing because it is seemingly used at various places for many disparate things as strategies and circuits. Then Figure 3 distinguishes between Mechanistic Interpretability versus Concept-based neologisms, which seems to contradict the more liberal use of "concept" throughout the manuscript.

__W7 (minor).__ The position might be more clearly stated if it includes the positive proposal (i.e., some form of "AI interpretability needs neologisms").

__W8 (minor).__ "Erring too high-level is alike to only doing behavioral testing of a network" (p. 2). Behavioral testing need not correspond to a particular level of abstraction, as behavioral experiments on cognitive systems have been used to gain mechanistic insights at various different levels, from cognitive modules to neural populations, in the discipline of Cognitive Science.

__W9 (minor).__ "...behavioral experiments/capability benchmarking" (p. 3; and Figure 3). Behavioral experiments are not equivalent to capability benchmarking. Cognitive Science has conducted behavioral experiments for decades, many of which are not aimed at benchmarking a capacity but rather at gaining insight on the mechanism behind a capacity or adjudicating between candidate models or a capacity.

__W10 (minor).__ "The words that survive to become neologisms strike a useful level of abstraction" (Citation needed?).

**Support:**

3

---

> ### Author Rebuttal · Authors · 2025-04-01
>
> Reviewer oDL4, thank you for your review! Your calls to engage more with relevant neuroscience-inspired and theoretical work in interpretability are well-taken, and we appreciate your evaluation of our work as “well-written” and as providing “extensive conceptual support.”
>
> **W1.** We appreciate the references in neuroscience-inspired interpretability and related fields that we can use to improve our work. We will include a more thorough discussion in our camera-ready, in particular referencing the articles you suggested on inner interpretability as well as Marr’s levels of analysis. Previously, we thought that our notions of abstraction did not align enough with Marr’s levels to prioritize comparison. In light of this review we will add such a comparison.
>
> **W2.** We will make more concrete our language in our camera-ready; in particular, in the quoted phrase “we still fall victim to our own biases...”, we will replace with “our biases still make it difficult for us to realize when we’ve not discovered a human concept like sentiment, but only a similar-seeming concept that differs in unpredictable ways."
>
> **W3.** Our claim that neologisms “provide a human interface for control” we will clarify as a proposal of our position and not a conclusion. We will further clarify that this is because using language is already used, e.g., in prompting image and video generation systems.
>
> **W4.** We’ll add citations for our claims about the continuing existence of adversarial examples and the utility of abstraction in understanding complex systems. Our statements of alternative positions, however, do not have specific citations, as they are synthesized from our understanding of AI and LLM development; instead we cite specific claims within those positions.
>
> **W5.** We will clarify in our new Figure 4 caption that the purple word “Ensure” is a new word learned via neologism embedding learning to enable a specific form of control.
>
> **W6.** Regarding our use of the word concept – we use concepts similar to those defined in Schut et al, 2023 [1]. We will include this definition in our camera-ready, along with our existing examples, e.g., language model safety, sentiment (of a text), move 37 in the AlphaGo series vs. Lee Sedol.
>
> [1] Schut, L., et al. Bridging the human-AI knowledge gap: Concept discovery and transfer in alphazero.
>
> **W7.** By stating the negative in our position, we hope to make the title more accessible before we discuss neologisms.
>
> **W8, W9.** We agree that behavioral testing can provide insight at different levels of Marr’s levels of analysis. We will clarify that we mean capability benchmarking here.
>
> **W10.** We will add citations that new words must be used frequently (thus not reference too specific a concept)  to spread in computational models of lexical change [2] as well as the notion of blocking [3] – that words don’t show up when there exist existing similar-enough words.
>
> [2] Eisenstein et al. Diffusion of Lexical Change in Social Media. PLoS One.
>
> [3] Aronoff (1976). Word Formation in Generative Grammar. MIT Press.
>
> Q1. _Are higher-level abstractions in this paper...more computationally tractable?..._
> - A1. Thank you for providing the references on the computational complexity of circuit discovery. We will reference them in our camera-ready to provide great evidence for the difficulty of circuit discovery. Whereas the utility of a circuit is in part its approximate minimality, which is computationally difficult to discover (e.g,. a bounded locally sufficient circuit), in learning neologisms, we are not attempting to discover a small circuit, but to learn a new word that implements a given behavior. We expect computational barriers -- we expect no guarantees that neologisms behave as expected globally. Empirically, however, we have high hopes for the direction.
>
> Q2. _Can you make the chess examples accessible to non-chess players?_
>  - A2. Certainly! In our camera-ready, we will give brief descriptions of these chess concepts.
>
> Q3. _Can you clarify the reason why "model outputs" are higher in abstraction level than circuits and concepts?..._
>  - A3. Thank you for this question. By abstraction, we mean mathematical abstraction – the stripping away of details. For example, stating a function y=f(x) abstracts away the structure of f. Stating it as or y=mx+b, is low-abstraction, stating the exact function. Stating or “y = f(x) where f is convex”, provides an intermediate level of abstraction. We do not mean abstraction in the sense of Marr. So, analyzing only the outputs of the model (e.g., capability benchmarking), one does not look at any internal component (high mathematical abstraction.) At the other end is using the entire computation graph – all neurons – in an experiment (e.g., mechanistic interpretability). In the middle is where we believe concepts fit in – in our case, learned and stored in new words – this abstracts over individual neurons and circuits. We will discuss in our camera-ready.

---

> > ### Comment · Reviewer_oDL4 · 2025-04-02
> >
> > Thanks for your response. All proposed changes seem adequate. I would gladly raise my score on the basis of concrete changes or new revisions of the manuscript but they don't seem to have been implemented yet.

---

> > > ### Author Response · Authors · 2025-04-02
> > >
> > > Dear Reviewer, oDL4,
> > >
> > > Thanks for your prompt response. We're happy to hear that the proposed changes seem adequate and that you'd be willing to raise your score!
> > >
> > > As much as we would have loved to update the manuscript directly, this is unfortunately neither allowed nor possible in ICML's OpenReview setup until the camera ready version. Quoting from ICML's email to authors, "No submission updates: Similar to previous years, the original submission (PDF and supplemental material) cannot be revised in OpenReview during the discussion period."
> > >
> > > We do, however, commit to incorporating the proposed changes exactly as described in our rebuttal as soon as we're allowed to upload a revision.

---

### Official Review · Reviewer_EqK1 · 2025-03-14

**Significance:** 3
**Argument Clarity:** 4
**Rating:** 5
**Confidence:** 4

**Questions:**

In reference to my comment about weaknesses, how do you think this call for AI neologisms relates to, or is (not) similar to, the creation of a new scientific field, and its corresponding jargon?
Is it broader than just a scientific field?

**Discussion Potential:**

3

**Paper Summary:**

Paper position: to understand AI we need to develop new words (neologisms)

The authors explain the importance of communication between AI and Humans as key factor to control and understand AI.
New words fill in a communication gap at a desired level of abstraction, enable compositionality, and help mitigate confirmation biases.
In Section 2, the paper makes the case for the need of neologisms on the basis that "humans and machines understand the world differently.". They then provide an example of how new words help share new and complex concepts, concisely refer to new learnable concepts, help strike a useful level of abstraction improving interpretability, and lessen confirmation biases. [examples of chess players]
In Section 3, there is a discussion about alternative views, such as 1) Differences in ways human and machine understand will disappears with time; 2) Existing vocabulary is sufficient; 3) All we need is a comprehensive map of models; 4)Mechanictic understanding does not need abstraction. Authors provide a rebuttal for each alternative position.
In Section 4, authors discuss how neologisms can connect to current interpretability models.
Finally, in Section 5, they discuss a proof of concept on a "diversity" neologism, tested on a specific example.


## update after rebuttal
I have read the rebuttal, and I confirm my ratings.

**Position:**

Yes

**Position In Title:**

Yes

**Related Work:**

4

**Strengths And Weaknesses:**

Strenghts

The paper supports the stated position with a resonably wide analysis.
The reasoning and evidence provided are convincing.
The "alternative views" part has a well done rebuttal subsection.
The proof of concept is important, and in my opinion fundamental, to explain/understand the core concept of the paper.

Weaknesses

This is not a critical weakness, as it may be due to limited space.
As this approach could be, and has been, used in different fields, there is a missing connection with previous epochal technological leaps.
For instance, physicists had to create new words (and new math: probability and functional analysis) to understand quantum mechanics.

**Support:**

4

---

> ### Author Rebuttal · Authors · 2025-04-01
>
> Reviewer EqK1, Thank you for your review! We’re glad to hear you’re strongly in favor of acceptance, given the paper’s “convincing reasoning and evidence” and an “important & fundamental proof of concept”.
>
> _Connection with previous epochal technological leaps:_
>
> Thanks for pointing out this fascinating connection between neologisms (i.e. creating new words) and technological progress. We’ll add a discussion on this point to the camera ready version (which has an extra page according to ICML). Broadly speaking, there are two complementary forms of influence between neologisms and technological progress:
> 1.  **New words/concepts arise from technological progress.** As technology advances, there is a need for new words to describe it - whether it’s a term for an invention, or a mathematical concept created to better understand a quantum system, or descriptive terminology for a newly discovered type of plant.
> 2. **New words/concepts can drive technological progress.** Conversely, while many words and concepts are created for a specific purpose or need, over time they themselves can become drivers of progress well beyond that original purpose. For instance, Newton and Leibniz are credited with the invention of differential equations. Today, differential equations are fundamental underpinnings of science and technology, including for instance rocket launches, neuroscience and submarine hydrodynamics —none of which were ever on Newton and Leibniz’ mind when developing the concept in the first place. Neologisms, thus, often enable downstream progress.
>
> As a result, neologisms are deeply tied to technological leaps, both as a result from progress and as a key enabler of progress. Similarly, our hope is that creating human-accessible neologisms for machine concepts may drive progress in interacting with AI and controlling it better.

---

### Official Review · Reviewer_BvYj · 2025-03-14

**Significance:** 2
**Argument Clarity:** 3
**Rating:** 3
**Confidence:** 3

**Questions:**

1. Have you considered testing the method on larger models or more complex tasks? What challenges might arise?
2. Can you provide additional evidence or references (e.g., from cognitive science) to support the claim of fundamental differences in human-machine conceptualization?

**Discussion Potential:**

3

**Paper Summary:**

This position paper proposes a novel approach to understanding and controlling AI systems by creating neologisms—new words—to bridge the conceptual gap between humans and machines. The authors frame AI interpretability as a communication problem, arguing that existing vocabulary is insufficient due to differing conceptualizations. They introduce "neologism embedding learning" and demonstrate its efficacy through experiments on the Gemma 2B model with the LIMA dataset.

"## update after rebuttal"
Thanks for the authors' rebuttal. I think they solved my concerns, thus I maintain my score.

**Position:**

Yes

**Position In Title:**

Yes

**Related Work:**

3

**Strengths And Weaknesses:**

Strengths
1. Framing AI interpretability as a communication problem and proposing neologisms as a solution is a fresh and interdisciplinary approach, drawing on linguistics, cognitive science, and AI.
2. The "neologism embedding learning" method is a concrete step toward implementing the proposed idea, preserving the original model's behavior when new words are not used (as shown in Figure 4).
3. The experiments on length control, diversity, and model preferences provide a proof-of-concept, showing how neologisms can be used to control and understand LLMs in a lightweight manner.

Weaknesses
1. The paper heavily relies on the premise of human-machine conceptual differences but does not sufficiently explore or justify this assumption.
2. The experiments are conducted on a small model (Gemma 2B) with a single dataset (LIMA), limiting the generalizability of the method. Larger models and diverse tasks (e.g., reasoning, multimodal tasks) should be included.
3. Some sections are verbose and lack precision. For example, the discussion of abstraction levels (Section 2.1) is abstract itself and would benefit from more concrete examples or formal definitions.

**Support:**

3

---

> ### Author Rebuttal · Authors · 2025-04-01
>
> Reviewer BvYj, thank you for your review! we’re happy to hear you appreciated the “fresh and interdisciplinary approach, drawing on linguistics, cognitive science, and AI” and the fact that the method “preserves the original model's behavior when new words are not used”.
>
> Q. _Can you provide additional evidence or references (e.g., from cognitive science) to support the claim of fundamental differences in human-machine conceptualization?_
>  - A. That’s a fair point, and we’re happy to substantially expand our discussion of this (fortunately, the camera ready version allows for an extra page) and include references from the literature to support this important assumption! A comprehensive overview could probably fill a review paper, but here are some references for human-machine differences that we will add to substantiate our argument, across language and vision; we welcome suggestions should you have additional references in mind.
>
> In language:
>  - [Abstract Understanding of Core-Knowledge Concepts: Humans vs. LLMs](https://openreview.net/pdf?id=bFWBD4UvUk)
>  - [Challenging the Validity of Personality Tests for Large Language Models](https://tomsuehr.com/wp-content/uploads/2024/06/challenging_the_validity_of_personality_tests_on_llms.pdf)
>  - [Can large language models generalize analogy solving like children can?](https://arxiv.org/pdf/2411.02348?)
>  - [Testing the limits of natural language models for predicting human language judgments](https://arxiv.org/pdf/2204.03592)
>
> In vision:
>  - [Shortcut Learning in Deep Neural Networks](https://www.nature.com/articles/s42256-020-00257-z)
>  - [Deep problems with neural network models of human vision](https://www.cambridge.org/core/journals/behavioral-and-brain-sciences/article/deep-problems-with-neural-network-models-of-human-vision/ABCE483EE95E80315058BB262DCA26A9)
>  - [ImageNet-trained CNNs are biased towards texture; increasing shape bias improves accuracy and robustness](https://openreview.net/forum?id=Bygh9j09KX&trk=public_post_comment-text)
>  - [Are Deep Neural Networks Adequate Behavioral Models of Human Visual Perception?](https://www.annualreviews.org/content/journals/10.1146/annurev-vision-120522-031739)
>  - [Explaining and harnessing adversarial examples](https://arxiv.org/abs/1412.6572)
>  - [ARC prize](https://arcprize.org/)
>  - [A comparison between humans and AI at recognizing objects in unusual poses](https://arxiv.org/abs/2402.03973)
>
> Q. _Larger models and diverse tasks (e.g., reasoning, multimodal tasks) should be included / What challenges might arise?_
>  - A. We fully agree, and we’re keen to explore this further. Given the nature of a position paper we kept the experimental setup simple to demonstrate a proof-of-concept, but we’re hoping to inspire follow-up experimental work with a broader set of models and tasks, including reasoning / “thinking” models as well as tackle some of those questions ourselves.
>
> Q. _The discussion of abstraction levels (Section 2.1) is abstract itself and would benefit from more concrete examples or formal definitions:_
>  - A. Thanks for flagging this; we will make Section 2.1 more specific and tangible.
>
> Thanks again for the feedback, and kindly let us know if this addresses your points, including whether you feel that the listed references sufficiently explore/justify the assumption of human-machine conceptual differences.

---

### Official Review · Reviewer_QeB2 · 2025-03-17

**Significance:** 4
**Argument Clarity:** 3
**Rating:** 4
**Confidence:** 3

**Questions:**

can we learn from naturally emergig neologisms in the recent past (like doom scroll)  and use that to understand which neologisms (H --> M) are needed.

**Discussion Potential:**

4

**Paper Summary:**

The paper posits that new words need to created for machines to better "understand" humans and vice-versa. The  technical word is to develop "neologisms." As a concrete example they use move 37 in AlphaGO which was a move that surprised experts but then was crucial for AlphaGO in winning the game. In Section 5, the authors provide concrete examples of inventing neologisms for specific LLM tasks like controlling response length and diversity which typically are hard for LLMs to follow with existing words.

**Position:**

Yes

**Position In Title:**

Yes

**Related Work:**

3

**Strengths And Weaknesses:**

The paper is very well written (though there a few minor typos) and clearly explains the main idea in a concrete manner.  The use of conceptualization at various levels, abstraction. The concrete example of concepts in GO and also how in Chess neologisms are used to train Chess players is very appropriate. I particularly liked Section 5, where they actually do a proof of concept of creating new words and imbuing them with precise meaning for tasks related to LLMs.

As a weakness, though they briefly mention, I think the authors should have elaborated on "adversarial prompts" a bit more and also related neologisms to manage different forms of uncertainty (e.g. epistemic vs aleatoric).

**Support:**

3

---

> ### Author Rebuttal · Authors · 2025-04-01
>
> Reviewer QeB2, thank you for your review! We particularly appreciate your evaluation of the work as “very well written,” and your positive evaluation of our proof-of-concept experiments to concretize and support our position. For the camera ready version, we will  improve our discussion of adversarial examples, and discuss how neologisms can help understand a model’s epistemic vs aleatoric uncertainties.

---

### Decision · Program_Chairs · 2025-04-30

**Decision:**

Accept (spotlight poster)

**Comment:**

The reviewers praised the position defended in the paper and unanimously recommended acceptance: 2 weak accept, 1 accept, 1 strong accept. The authors have responded in detail to the reviewers' comments and questions and I believe that the discussion will contribute to improving the quality of the final manuscript.